# Toll-Like Receptors-2 and -4 in Graves’ Disease—Key Players or Bystanders?

**DOI:** 10.3390/ijms20194732

**Published:** 2019-09-24

**Authors:** Agnieszka Polak, Ewelina Grywalska, Janusz Klatka, Jacek Roliński, Beata Matyjaszek-Matuszek, Maria Klatka

**Affiliations:** 1Department of Endocrinology, Medical University of Lublin, 20-954 Lublin, Poland; beata.matyjaszek_matuszek@wp.pl; 2Department of Clinical Immunology and Immunotherapy, Medical University of Lublin, 20-093 Lublin, Poland; ewelina.grywalska@gmail.com (E.G.); jacek.rolinski@gmail.com (J.R.); 3St. John’s Cancer Centre, Jaczewskiego 7 Street, 20-090 Lublin, Poland; 4Department of Otolaryngology and Laryngeal Oncology, Medical University of Lublin, 20-954 Lublin, Poland; klatka.janusz@wp.pl; 5Department of Pediatric Endocrinology and Diabetology, Medical University, 20-093 Lublin, Poland; klatka_maria@wp.pl

**Keywords:** Graves’ disease, Toll-like receptor 2, Toll-like receptor 4, T lymphocytes, B lymphocytes, cytokines

## Abstract

Graves’ disease (GD) is an autoimmune disease that affects the thyroid. The development of autoimmunity is associated with innate immune responses where the prominent role plays Toll-like receptors (TLRs). The aim of our study was to assess the relationship between the expression levels of TLR-2 and TLR-4 on CD4+ and CD8+ T as well as CD19+ B lymphocytes in patients with GD and selected clinical parameters. The study group consisted of 32 women with GD, the control group consisted of 20 healthy women. Immunophenotyping was performed using the flow cytometry and cytokines concentrations were assessed using ELISA assay. The mean percentage of CD4+/TLR-2+ and CD8+/TLR-2+ T cells in patients with GD was higher than in the control group (*p* < 0.0001). After obtaining euthyroidism, the mean percentage of CD4+/TLR-2+ T cells in patients with GD decreased (*p* < 0.0001). The expression level of TLR-2 on CD4+ T lymphocytes correlated with serum FT3 concentration in patients with GD (*r* = 0.47, *p* = 0.007). The mean percentage of CD8+/TLR-2+ T cells in patients with GD before treatment compared to patients with GD after obtaining euthyroidism was higher (*p* = 0.0163). Similar findings were found for TLR-4. Thus the TLR-2 and TLR-4 can be a prognostic marker for Graves’ disease.

## 1. Introduction

Graves’ disease (GD) is an autoimmune thyroid disease with complex and incompletely established etiology. It may occur in genetically predisposed individuals who are exposed to certain exogenous factors, such as iodine, selenium, smoking, bacterial and viral infections, allergies, radiotherapy or emotional stress. Endogenous factors such as age, sex, pregnancy, hormones and birth weight are also important [1,2,3,4]. GD develops due to a loss of immunological tolerance and reactivity to thyroid autoantigens, mainly directed against the thyrotropin receptor. This leads to the infiltration of this gland by T lymphocytes and B lymphocytes that produce antibodies specific for this disease entity, with a clinical manifestation of hyperthyroidism in the form of GD [1,2,3,4]. Recent studies have confirmed the importance of toll-like receptors (TLRs) and proinflammatory cytokines in the etiopathogenesis of autoimmune diseases. In GD, the thyroid gland grows as a result of the stimulation of follicular cells by anti-receptor antibodies (TRAb). In view of the disturbing increase in the incidence rate of GD, hyperthyroidism with an autoimmune background has become an important health problem [5]. Cytokines involved in the pathomechanism of GD influence both the functioning of the immune system of the whole body of the ill person as well as directly on the targeted thyroid follicular cells. They participate in the induction and effector phase of the immune and inflammatory response, thus playing a key role in the pathogenesis of the disease. Within inflammatory cells and the thyroid, follicular cells increasing expression of many cytokines has been detected, including interleukin (IL)-1 alpha, IL-1 beta, IL-2, IL-4, IL-6, IL-8, IL-10, IL-12, IL-13, IL-14, tumor necrosis factor alpha (TNF-alpha) and interferon gamma (IFN-gamma) [1,3,4,5]. Studies have shown that gene polymorphisms for cytokines or their abnormal secretion may not only increase the risk of developing GD but may also affect the course of the disease and prognosis [6]. It has been found that thyrocytes in patients with GD do not undergo apoptosis, and in their vicinity, there is infiltration consisting mainly of Th2 cells, which secrete small amounts of IL-2, TNF-alpha and IFN-gamma. However, they are capable of producing large amounts of IL-4, IL-5 and IL-10 [7].

In this study, the relationship between TLR-2 and TLR-4 expression on immune cells and the concentration of selected T helper (Th)1 and Th2 cytokines in peripheral blood of women with GD before thyrostatic treatment and after receiving euthyroidism was assessed. Moreover, the effect of the above parameters on selected clinical variables was evaluated.

## 2. Results

### 2.1. Comparison of the Percentage Values of CD4+/TLR-2+ T Cells in GD Patients before Treatment, after Treatment and in the Control Group

The mean percentage values of CD4+/TLR-2+ T cells in the studied groups of patients were 1.32% ± 0.59% in patients with GD before treatment, 0.26% ± 0.11% in the control group and 0.59% ± 0.09% in patients with GD after treatment (Appendix A, Figure 1). The analysis of variance showed that the mean percentage of CD4+/TLR-2+ T cells in GD patients in comparison with the control group was statistically significantly higher (*p* < 0.0001; Figure 1). After obtaining euthyroidism as a result of treatment, the mean percentage of CD4+/TLR-2+ T cells in patients with GD decreased, and this difference was statistically significant compared to the values before treatment (*p* < 0.0001; Figure 1). The mean percentage of CD4+/TLR-2+ T cells in patients with GD after obtaining euthyroidism in comparison to the control group was statistically significantly higher (*p* = 0.0003; Figure 1).

### 2.2. Evaluation of TLR-2 Expression Level on CD4+ T Cells in GD Patients before Treatment, after Treatment and in the Control Group

Analysis of variance showed that there were no statistically significant differences in the expression of TLR-2 on CD4+ T lymphocytes in the examined groups of patients (Appendix A).

### 2.3. Comparison of the Percentage Values of CD8+/TLR-2+ T Cells in Patients with GD before Treatment, after Treatment and in the Control Group

The mean percentage values of CD8+/TLR-2+ T cells in the examined groups of patients were 4.01% ± 2.37% in patients with GD, 1.30% ± 1.46% in the control group and 2.35% ± 1.31% in patients with GD after treatment (Appendix A). The analysis of variance showed that the mean percentage of CD8+/TLR-2+ T cells in GD patients in comparison with the control group was statistically significantly higher (*p* < 0.0001; Figure 2). The mean percentage of CD8+/TLR-2+ T cells in patients with GD before treatment compared to the patients with GD after obtaining euthyroidism was statistically significantly higher (*p* = 0.0163; Figure 2).

### 2.4. Evaluation of TLR-2 Expression Level on CD8+ T Cells in GD Patients before Treatment, after Treatment and in the Control Group

Differences in the expression level of TLR-2 on CD8+ T lymphocytes in the examined groups of patients were observed. The MFI (mean fluorescence intensity) of the TLR-2 on CD8+ T cells was 54.54 ± 52.25 in GD patients before treatment, 33.37 ± 43.23 in the control group and 44.90 ± 35.62 in patients with GD after treatment (Appendix A). The analysis of variance showed that the MFI of the TLR-2 on CD8+ T cells in GD patients in comparison with the control group was statistically significantly higher (*p* = 0.0054; Figure 3). The mean expression level of CD8+/TLR-2+ T cells in patients with GD after obtaining euthyroidism in comparison to the control group was statistically significantly higher (*p* = 0.0481; Figure 3).

### 2.5. Comparison of the Percentage Values of CD19+/TLR-2+ B Cells in Patients with GD before Treatment, after Treatment and in the Control Group

The mean percentage values of CD19+/TLR-2+ B cells in the examined groups of patients were 1.54% ± 1.04% in patients with GD before treatment, 0.88% ± 0.86% in the control group and 1.32% ± 1.33% in patients with GD after treatment (Appendix A). The analysis of variance showed that the mean percentage of CD19+/TLR-2+ B cells in GD patients in comparison with the control group was statistically significantly higher (*p* = 0.0127; Figure 4).

### 2.6. Evaluation of TLR-2 Expression Levels on CD19+ B Cells in GD Patients before Treatment, after Treatment and in the Control Group

Analysis of variance showed that there were no statistically significant differences in the expression of TLR-2 on CD19+ B lymphocytes in the examined groups of patients (Appendix A).

### 2.7. Comparison of the Percentage Values of CD4+/TLR-4+ T Cells in Patients with GD before Treatment, after Treatment and in the Control Group

The mean percentage values of CD4+/TLR-4+ T cells in the studied groups of patients were 0.63% ± 0.23% in patients with GD before treatment, 0.24% ± 0.25% in the control group and 0.57% ± 0.23% in patients with GD after treatment (Appendix A). The analysis of variance showed that the mean percentage of CD4+/TLR-4+ T cells in patients with GD in comparison with the control group was statistically significantly higher (*p* < 0.0001; Figure 5). The mean percentage of CD4+/TLR-4+ T cells patients with GD after obtaining euthyroidism in comparison to the control group was statistically significantly higher (*p* = 0.0003; Figure 5). There were no statistically significant differences in mean percentage values of CD4+/TLR-4+ T cells patients with GD before treatment in comparison with patients with GD after obtaining euthyroidism.

### 2.8. Evaluation of TLR-4 Expression Level on CD4+ T Cells in GD Patients before Treatment, after Treatment and with the Control Group

The mean percentage of CD8+/TLR-4+ T lymphocytes was 2.74% ± 1.07% in patients with GD before treatment, 0.98% ± 1.24% in the control group and 2.55% ± 1.50% in patients with GD after treatment (Appendix A). The analysis of variance showed that the mean percentage of CD8+/TLR-4+ T cells in patients with GD in comparison with the control group was statistically significantly higher (*p* < 0.0001; Figure 6). After obtaining euthyroidism as a result of the treatment, the mean percentage of CD8+/TLR-4+ T cells in patients with GD decreased and this difference was statistically significant in comparison with the control group (*p* < 0.0001; Figure 6). The mean percentage of CD8+/TLR-4+ T cells in patients with GD after euthyroidism in comparison with the control group was statistically significantly higher (*p* = 0.0003; Figure 6).

### 2.9. Evaluation of TLR-4 Expression Level on CD8+ T Cells in GD Patients before Treatment, after treaTment and in the Control Group

Differences in the expression level of TLR-4 on CD8+ T lymphocytes in the studied group of patients were observed (Appendix A). The MFI of TLR-4 on CD8+ T cells in patients with GD before treatment was 158.92 ± 145.89 and 133.96 ± 54.35 after treatment. The MFI of these receptors in the control group was 73.38 ± 109.67 (Appendix A). The analysis of variance showed that MFI of TLR-4 on CD8+ T lymphocytes in patients with GD in comparison with the control group was statistically significantly higher (*p* = 0.0075; Figure 7). The MFI of CD8+/TLR-4+ T cells in patients with GD after euthyroidism in comparison with the control group was statistically significantly higher (*p* = 0.0023; Figure 7).

### 2.10. Comparison of the Percentage Values of CD19+/TLR-4+ B Cells in Patients with GD before Treatment, after Treatment and in the Control Group

The mean percentage values of CD19+/TLR-4+ B lymphocytes in patients with GD before treatment was 1.68% ± 1.84% and 1.32% ± 1.39% after treatment. The mean percentage value of these cells in the control group was 0.69% ± 0.78% (Appendix A). The analysis of variance showed that the mean percentage value of CD19+/TLR-4+ B lymphocytes in patients with GD in comparison with the control group was statistically significantly higher (*p* = 0.0361; Figure 8).

### 2.11. Evaluation of TLR-4 Expression Level on CD19+ B Cells in GD Patients before Treatment, after Treatment and in the Control Group

Analysis of variance showed there were no statistically significant differences in the expression level of CD19+/TLR-4+ B cells in patients with GD before treatment in comparison with patients with GD after obtaining euthyroidism and control group (Appendix A).

### 2.12. Evaluation of Correlations between the Expression of the Analyzed TLRs on Particular Cell Populations and Clinical Parameters in Patients with GD Prior to the Implementation of Antithyroid Therapy, after Obtaining Euthyroidism and in the Control Group

A statistically significant correlation was shown between the expression level of CD4+/TLR-2+ T lymphocytes and CD4+/TLR-4+ T lymphocytes in patients with GD before treatment (Figure 9).

A statistically significant correlation was found between the expression level of TLR-2 on CD8+ T lymphocytes and the expression level of TLR-4 on CD8+ T lymphocytes (Figure 10) and between the expression level of TLR-2 on CD19+B lymphocytes and the expression level of TLR-4 on CD19+ B lymphocytes in patients with GD before treatment (Figure 11).

A statistically significant correlation was also found between the expression level of TRL-2 on CD8+ T lymphocytes and the expression level of TRL-2 on CD4+ T lymphocytes in patients with GD after treatment (Figure 12).

A statistically significant correlation was shown between the expression level of TLR-2 on CD 19+ B lymphocytes and the expression level of TLR-4 on CD4+ T lymphocytes (Figure 13) and between the expression level of TLR-2 on CD19+ B lymphocytes and the expression level of TLR-4 on CD19+ B lymphocytes (Figure 14) in patients with GD after treatment and obtaining euthyroidism.

A statistically significant correlation was found between the expression level of TLR-2 on CD4+ T lymphocytes and FT3 concentration in patients with GD before treatment (Figure 15). The expression of TLR-2 on CD4+ T lymphocytes did not correlate with TSH concentration (r = −0.06, *p* = 0.72), FT4 concentration (r = 0.19, *p* = 0.29), sum activity of peripheral deiodinases (SPINA-GD; r = 0.11, *p* = 0.54), thyroid’s secretory capacity (SPINA-GT; r = −0.13, *p* = 0.47), Jostel’s THSK Index (JTI; r = 0.23, *p* = 0.19), TRAb concentration (r = −0.11, *p* = 0.53) or thyroid volume (r = −0.03, *p* = 0.84).

A statistically significant correlation was also found between the expression of TLR-4 on CD19+ B lymphocytes and FT3 concentration in patients with GD before treatment (Figure 16). The expression of TLR-4 on CD19+ B lymphocytes did not correlate with TSH concentration (r = −0.10, *p* = 0.55), FT4 concentration (r = 0.11, *p* = 0.52), SPINA-GD (r = 0.27, *p* = 0.12), SPINA-GT (r = −0.10, *p* = 0.55), JTI (r = 0.153, *p* = 0.40), TRAb concentration (r = −0.06, *p* = 0.71) or thyroid volume (r = −0.21, *p* = 0.24).

### 2.13. Evaluation of the Sum Activity of Peripheral Deiodinases (SPINA-GD), Thyroid’s Secretory Capacity (SPINA-GT) and the Jostel’s THSK Index (JTI)

Compared to controls, SPINA-GD was significantly increased in patients with GD before treatment, and it decreased to near normal values when the patients became euthyroid (Table 2). Similarly, the thyroid secretory capacity (SPINA-GT) was markedly increased in patients with GD before treatment, and it decreased, but did not normalize, when the patients were euthyroid (Table 2). The mean JTI was negative in patients with GD, and it decreased further after thyrostatic treatment (Table 2).

## 3. Discussion

Autoimmune diseases are characterized by overactive T lymphocytes and excessive stimulation of B lymphocytes, leading to increased production of autoantibodies [8]. TLRs are highly specialized receptors recognizing molecular patterns associated with pathogens [9]. They play a major role in the induction and regulation of both innate and acquired immune responses. These receptors are expressed in many cells of the immune system, including on macrophages, dendritic cells, mast cells, eosinophils, T and B lymphocyte subpopulations and on airway and gastrointestinal epithelial cells [9,10]. Thus, by initiating the signaling cascade, they initiate the expression of co-stimulatory molecules and the secretion of pro-inflammatory cytokines, i.e., the body’s defensive reaction, leading to the elimination of pathogens [11,12]. It is believed that TLRs are an element of the immune response that combines mechanisms of innate immunity and factors leading to the development of auto-aggressive diseases [13,14]. Signal transduction through TLRs occurs through the involvement of many proteins, such as MyD88 (myeloid differentiation factor 88), TAK1 kinases (TGF-beta-activated kinase), TAK1 kinase-binding proteins (TAK1-binding proteins), IRAK kinases ((IL-1)-1R1-associated protein kinases) and factor associated with the tumor necrosis factor receptor—TRAF6 (TNF-receptor-associated factor 6) [15]. Under certain circumstances TLR ligation may lead to abnormal activation and inflammatory reactions, thus contributing to the fixation of inflammation in autoimmune diseases. Most studies have concerned intra-cellular TLRs, such as TLR-3, TLR-7 and TLR-9, but recent studies have also shown that cell-surface TLRs, especially TLR-2 and TLR-4, play an equally important role in the development of autoimmune diseases [16]; however, relevant studies have not been conducted so far in women with GD.

One of the examples of autoimmune diseases is rheumatoid arthritis (RA). In their work, Yu Liu et al. demonstrated that TLR-2 and TLR-4 are expressed on macrophages and fibroblasts in RA patients. A higher level of expression of these receptors was observed in RA patients compared to healthy controls and in comparison to patients with osteoarthritis whose levels of these receptors were lower [17]. Another example is systemic lupus erythematosus (SLE). It is an autoimmune disease that leads to the inflammatory process of many tissues and organs. It is believed that SLE results from an interaction between genetic and environmental factors [18]. One of the mechanisms by which bacteria and viruses can participate in autoimmune disorders is their interaction with TLRs. Studies by Komatsud et al. indicated that TLR-2 mRNA levels significantly increased in the peripheral blood of SLE patients compared to the control group, whereas Kirchner et al. demonstrated that the level of TLR-4 expression on CD14 + monocytes was significantly lower in patients with SLE compared to the control group [19,20]. Another example of the role of TLRs in autoimmune diseases is vitiligo [21]. TLR expression was also detected in keratinocytes, melanocytes and Langerhans cells in the skin in many other dermatological diseases such as atopic dermatitis, psoriasis and acne vulgaris [22,23,24]. TLR-4 receptors in melanocytes can react with endogenous heat shock proteins and lead to the initiation of an autoimmune reaction [25]. In contrast, Devarai et al. demonstrated a significant increase in the expression of TLR-2 and TLR-4 in patients with type 1 diabetes compared to the control group [26]. There is also scientific evidence for the involvement of TLR in the pathogenesis of Sjogren’s syndrome. Kawakami et al. demonstrated that TLR-2 and TLR-4 receptors in the salivary gland of the disease patients show a significantly higher level of expression than in the control group [27]. Studies of recent years indicate that abnormal activation of the innate immune response, and consequently a number of changes resulting from this fact, may have a significant impact on the pathogenesis of autoimmune thyroid diseases (AITD) [28,29,30,31,32,33,34,35,36,37,38,39,40]. In light of the above data, TLRs play an important role in the development of autoimmune diseases such as SLE, type 1 diabetes or RA [17,18,19,20,21,22,23,24,25,26,27,28].

In the available literature, however, there are few data on the role of TLR in AITD, including GD disease. McGrogan et al. assume that the basic mechanisms involved in the development of inflammation in these diseases are the result of the action of adipokines, in particular, leptin, TNF-alpha and IL-6, as well as pathogen-associated molecular patterns (PAMP), damage-associated molecular patterns (DAMP) molecules and TLRs [41]. Adipokines are produced and secreted by adipose tissue cells (adipocytes). They exhibit autocrine, paracrine and endocrine properties on tissues and organs. A significantly higher level of leptin in women explains the difference in the incidence of autoimmune thyroid disease in both sexes [42].

Thyrocytes have the ability to participate in non-specific immunity via TLRs, which when bound to the relevant PAMP molecules transmit signals that activate cells [43].

Our research showed that the mean percentages of CD4+/TLR-2+ T cells, CD8+/TLR-2+ T cells and CD19+/TLR 2+ CD8 cells in peripheral blood were significantly greater in patients with GD than in the control group.

Similar results were obtained by Peng et al. They assessed the level of TLRs from 1–10 in patients with AITD. The study involved 66 patients with AITD (including 30 patients with newly diagnosed GD and 36 people with Hashimoto thyroiditis) and 30 healthy controls. The level of TLR 1–10 expression in the peripheral blood of patients was assessed by RT-PCR. They observed, among others things, a markedly increased TLR-2 expression in the peripheral blood of patients with newly diagnosed GD or with Hashimoto thyroiditis compared to the control group. Peng et al. were among the first researchers who showed an increased TLR expression in the peripheral blood of patients with AITD, which may support their participation in the pathogenesis of autoimmune thyroid diseases [44]. In addition, our own studies showed a statistically significantly higher percentage of CD4+ T cells, CD8+ T and CD19+ B cells expressing TLR-4 receptors in the peripheral blood of GD patients compared to their peripheral blood percentage in the control group. Available literature does not yet contain data on the assessment of the percentage of TLR-4+ lymphocytes in patients with GD, which makes it impossible to compare our results with previous studies in this disease entity. Similar results to the authors’ own findings presented above, however, were obtained by the authors analyzing TLR-4 expression in other autoimmune diseases, such as RA, psoriasis, type I diabetes and Sjogren’s syndrome, which confirms the participation of these receptors in the pathogenesis of autoimmune diseases [26,27,45].

In addition, based on our own studies, numerous statistically significant correlations were found between lymphocytes expressing TLRs and clinical parameters and a concentration of cytokines in the peripheral blood. In the group of patients with GD prior to treatment, there is a correlation between TLR-2 antigen expression on CD4+ T lymphocytes and TLR-4 antigen expression on CD4+ T lymphocytes, the correlation between TLR-2 antigen expression on CD4+ T lymphocytes and FT3 concentration. Pre-treatment patients also had a correlation between TLR-2 antigen expression on CD8 + T cells and TLR-4 antigen expression on CD8+ T cells and between TLR-2 antigen expression on CD19+ lymphocytes and TLR-4 antigen expression on CD19+ lymphocytes. Our own studies also showed a statistically significant correlation between the expression of TLR-4 antigen on CD19+ lymphocytes and peripheral blood FT3 in patients with GD before treatment.

In the peripheral blood of GD patients after thyrostatic therapy and receiving euthyroidism, statistically significant relationships were found between TLR-2 antigen expression on CD8+ T lymphocytes and TLR-2 antigen expression on CD4+ T lymphocytes. These patients also showed statistically significant correlations between TLR-2 antigen expression on CD19+ B lymphocytes and TLR-4 antigen expression on CD4+ T cells as well as TLR-2 antigen expression on CD19 + B lymphocytes and TLR-4 antigen expression on CD19+ lymphocytes B. It is difficult to explain the negative correlation between TLR2 expression on CD19+ cells and TLR4 expression on CD4+ cells. Perhaps, there is a negative feedback loop that controls the expressions of TLR-4 and TLR-2, but we are not aware of any previous evidence that could support this hypothesis.

In our study, SPINA-GD (sum activity of peripheral deiodinases) was increased in patients with GD before treatment, and it decreased to near normal values when the patients were euthyroid. SPINA-GD correlates with the FT4 to FT3 conversion rate; thus, an increased SPINA-GD may be related to the high-FT3 syndrome in GD. As one could expect, the thyroid secretory capacity (SPINA-GT) was markedly increased in patients with GD before treatment, and it decreased, but did not normalize, after thyrostatic treatment. The mean JTI was negative in patients with GD, which was due to very low TSH values [JTI = ln (TSH) + 0.1345 × FT4]. Interestingly, the JTI decreased further after thyrostatic treatment, likely because FT4 levels normalized faster than TSH in GD.

Our study was limited by a small sample; thus, other studies should replicate or extend our findings. Moreover, although our observational study points to the importance of TLRs in AITD, it cannot answer whether TLRs are key players or by-standers in the pathogenesis of GD. The role of TLRs in AITD should be further investigated in preclinical studies that enable TLR blockage or stimulation. Similarly, large studies among patients with GD could help reveal any link between TLRs expression and relevant clinical outcomes.

There is no data in the available literature describing the correlation between TLR-2 and TLR-4 and other factors elaborated in our study (i.e., thyroid hormone levels or concentration of selected cytokines) in GD. However, Garcia-Rodriguez et al. in their work on psoriasis patients found correlations between the expression of TLR-2 and IL-2, IL-4 and IL-10 levels. They showed that the percentages of TLR-2+ and TLR-4+ lymphocytes are higher in patients with psoriasis than in the general population. This confirms the thesis that TLR-2 and TLR-4 may be involved in the development of autoimmune diseases [45].

## 4. Materials and Methods

### 4.1. Characteristics of Study and Control Groups

The study group consisted of 32 women with newly diagnosed, previously untreated Graves’ disease, who were hospitalized in the Clinic of Endocrinology of the Medical University in Lublin. The average age of patients was 41 ± 16.21 years. The control group included 20 healthy women in the age of 39.3 ± 11.3 years, hospitalized in the Department of Otolaryngology, the Medical University of Lublin due to the distortion of the nasal septum and external nose deformity, with normal thyroid function, which was confirmed by hormonal tests. Both groups excluded patients taking medication affecting the immune system, reporting symptoms of infection in the last 12 weeks before the test, patients with diagnosed allergies, history of other concomitant autoimmune diseases and after a blood transfusion procedure. Neither the patients nor the controls used immunomodulating agents, vaccines or hormonal preparations; showed signs of infection within at least six months prior to the study; underwent blood transfusion; or presented with an allergy. Moreover, none of the patients and controls had a history of oncological therapy or prior treatment for an autoimmune condition or tuberculosis or other chronic conditions that could be associated with impaired cellular or humoral immunity.

Diagnosis of hyperthyroidism was based on the clinical features and laboratory exponents of hyperthyroidism (elevated FT3 and FT4 levels and decreased TSH level). GD was diagnosed on the basis of a positive anti-TSH receptor antibody (positive titer > 1.5 U/L). The research was carried out by radioimmunoassay at the Department of Radiology and Nuclear Medicine of the Medical University of Lublin. The mean period of thyrostatic therapy conducted among women constituting the study group from the moment of inclusion of the treatment to the normalization of the level of thyroid hormones and reduction of symptoms was about 6 weeks. The detailed characteristics of the patients and the control group are presented in Table 1 and Table 2.

Based on the structure parameter inference approach (SPINA), we calculated the sum activity of peripheral deiodinases (SPINA-GD), thyroid’s secretory capacity (SPINA-GT) and the Jostel’s THSK index (JTI). We used the SPINA package for the R software.

The study material was 5 mL of peripheral blood, collected in heparin-coated tubes to obtain peripheral blood cells, taken during routine laboratory tests. Each patient expressed written informed consent to participate in the study. The Bioethical Commission at the Medical University in Lublin (EC Resolution No. KE-0254/191/2018, the 29th January 2018) gave appropriate permission to conduct this research.

### 4.2. Assessment of the T CD4+, T CD8+ and B CD19+ Lymphocytes with the Expression Level of TLR-2 and TLR-4 Receptors

Peripheral blood was diluted with 0.9% buffered saline—PBS containing no calcium ions (Ca^2+^) and magnesium (Mg^2+^; Biochrome AG, Berlin, Germany) in a 1:1 ratio. The diluted material was layered on 3 mL Gradisol L (Aqua Medica, Lodz, Poland) with a specific gravity of 1.077 g/mL, and then centrifuged in a density gradient at 700× *g* for 20 min. The peripheral blood mononuclear cell fraction (PBMC) thus obtained was collected with Pasteur pipettes and washed twice in a PBS solution without Ca^2+^ and Mg^2+^ ions for 5 min. Subsequently, these cells were suspended in 1 mL of PBS containing no calcium and magnesium ions, and their numbers were determined in the Neubauer chamber and viability via trypan blue (0.4% Trypan Blue Solution, Sigma Aldrich, Saint Louis, USA).

Subsequently, obtained by PBMC density gradient centrifugation, 2 or 3-color labeling with monoclonal antibodies was used, which was used in an appropriate amount, as recommended by the manufacturer. The cell suspension obtained after isolation was split into individual tubes at 1 × 10^6^/sample and then incubated for 20 min at room temperature with the appropriate combination of the following monoclonal antibodies (20 μL antibody per sample): Anti-CD4 PerCP, anti-CD8 PE-Cy5, anti-CD19 PE-Cy5 (Becton Dickinson, San Diego, USA), anti-TLR2 FITC and anti-TLR4 FITC (Abcam, London, UK). After incubation, the cells were washed twice with physiological saline buffer (PBS; 700× *g*, 5 min) and then immediately analyzed in a FACSCalibur flow cytometer (Becton Dickinson, San Diego, USA). Data acquisition was performed using the FACS Diva Software 6.1.3 software, collecting 30,000 cells for each assay, while analyzing them using the CellQuest Pro program (Becton Dickinson, San Diego, USA). The results of the cytometric analysis were presented as a percentage of cells stained with monoclonal antibodies conjugated with fluorescent dyes and as mean fluorescence intensity (MFI), which is an exponent of the amount of expression of a given antigen on the cell surface. An example of the cytometric analysis of T CD4+ lymphocytes, T CD8+ lymphocytes and B CD19+ lymphocytes expressing TLR-2 or TLR-4 is shown in the figure below (Figure 17A–C).

### 4.3. Statistical Analysis

Statistical analysis was conducted with Statistica 10 PL software (StatSoft, United States). The fractions of identified cells and MFI values were expressed as means ± standard deviation (SD), medians and ranges. The Mann–Whitney U-test and Kruskal–Wallis test were used for intergroup comparisons. The differences were considered significant at *p* < 0.05. The power and direction of relationships between pairs of continuous variables were determined using Spearman’s coefficient of rank correlation.

## 5. Conclusions

Our study revealed statistically significant differences in the expression of TLR-2 and TLR-4 on T and B lymphocytes in patients with GD before the introduction of antithyroid therapy, after arriving at euthyroidism and in healthy subjects, which may suggest an important role in interactions mechanisms of non-specific and specific immunity in the pathogenesis of the disease. Numerous relationships between the expression of TLR-2 and TLR-4 on the surface of T lymphocytes (CD4 + and CD8 +) and B CD19 + in patients with GD and selected clinical and laboratory parameters indicate the important role of these receptors in the course of the disease. The evaluation of peripheral blood lymphocytes expressing TLR-2 and TLR-4 allows concluding that these parameters play an important role in etiopathogenesis as well as in the clinical course of GD disease. Further research should aim to define the factors underlying the interaction of specific and nonspecific components of the immune response, as well as the search for substances that will stimulate the immune system to improve the prognosis in cases resistant to standard treatment.

## Figures and Tables

**Figure 1 ijms-20-04732-f001:**
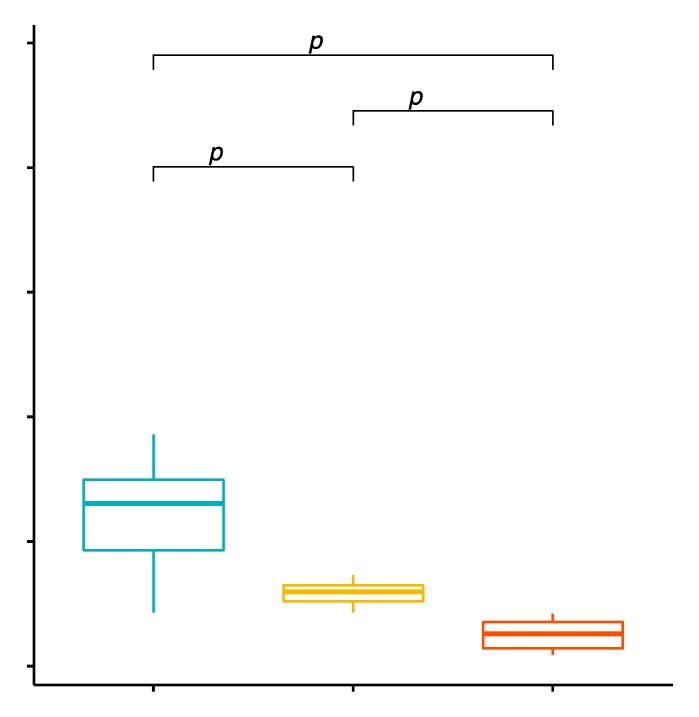
Percentages of CD4+/TLR-2+ T lymphocytes (among CD4+ cells) in patients with Graves’ disease (GD) before treatment, after euthyroidism and in the control group.

**Figure 2 ijms-20-04732-f002:**
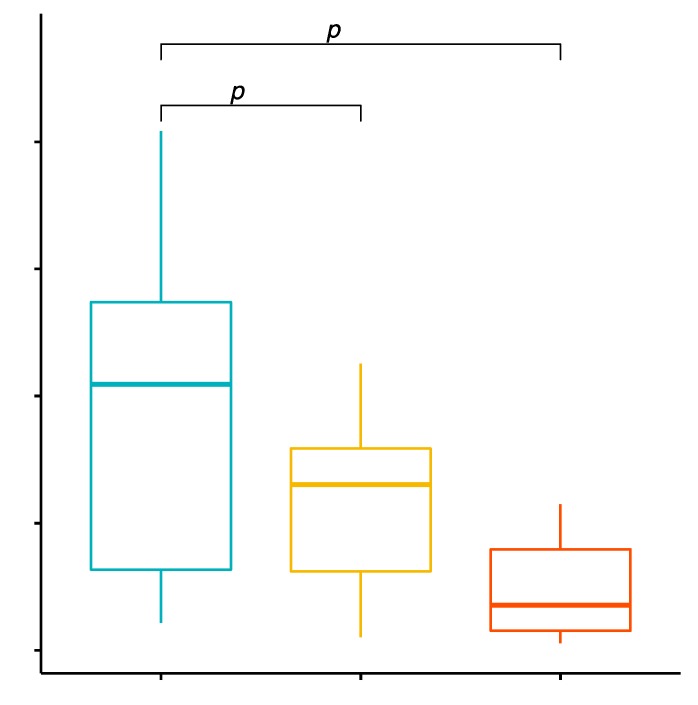
Percentage of CD8+/TLR-2+ T lymphocytes (among CD8+ cells) in patients with GD before treatment, after euthyroidism and in the control group.

**Figure 3 ijms-20-04732-f003:**
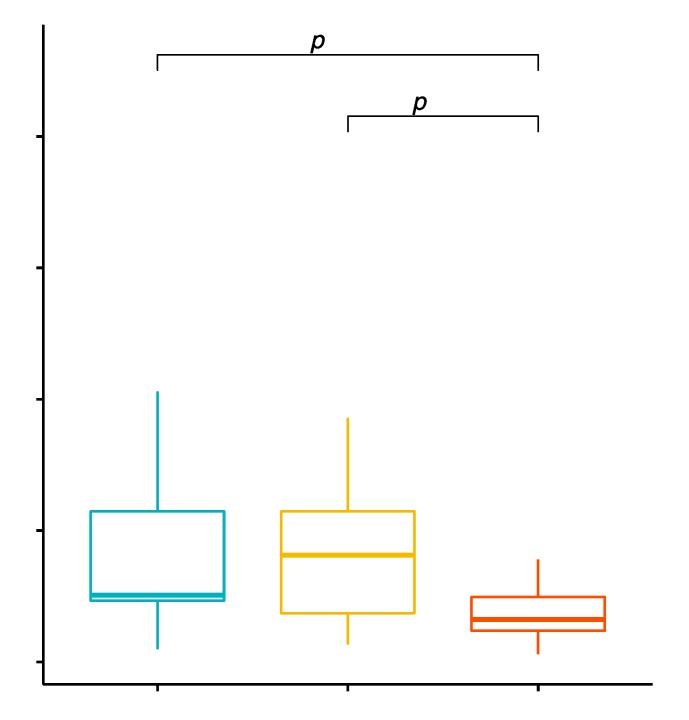
The expression level of TLR-2 on CD8+ T cells (mean fluorescence intensity—MFI) in patients with GD before treatment, after euthyroidism and in the control group.

**Figure 4 ijms-20-04732-f004:**
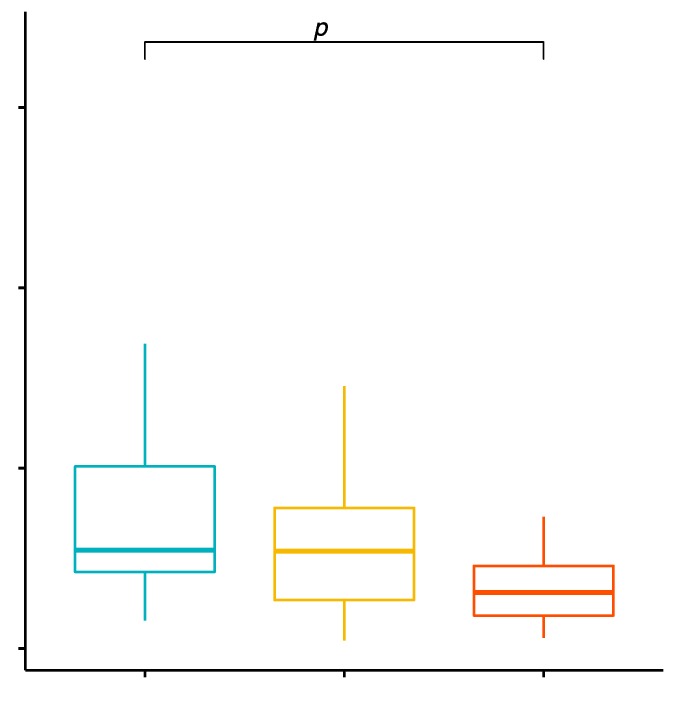
Percentage of CD19+/TLR-2+ B cells (among CD19+ cells) in patients with GD before treatment, after euthyroidism and in the control group.

**Figure 5 ijms-20-04732-f005:**
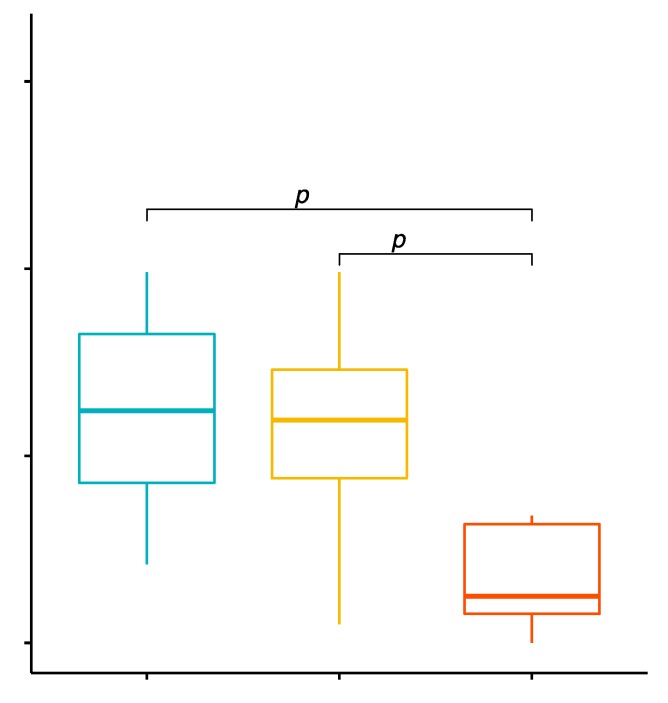
Percentage of CD4+/TLR-4+ T lymphocytes (among CD4+ cells; %) in patients with GD before treatment, after euthyroidism and in the control group.

**Figure 6 ijms-20-04732-f006:**
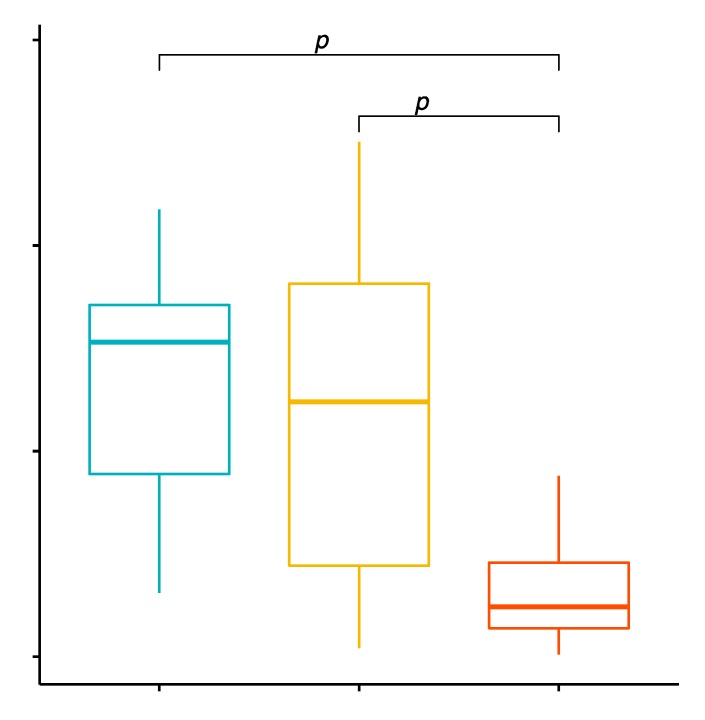
Percentage of CD8+/TLR-4+ T lymphocytes (among CD8+ cells; %) in patients with GD before treatment, after euthyroidism and in the control group.

**Figure 7 ijms-20-04732-f007:**
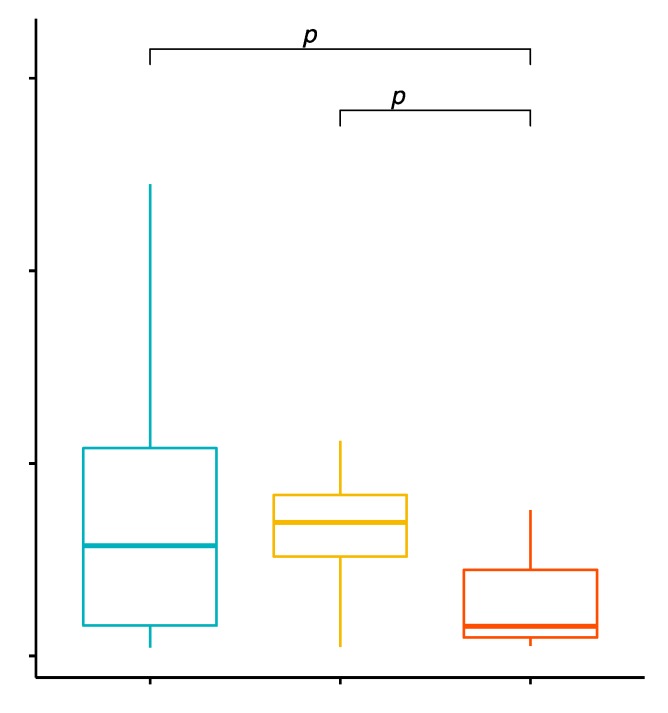
The expression level of TLR-4 on CD8+ T cells (MFI) in patients with GD before treatment, after euthyroidism and in the control group. MFI, mean fluorescence intensity.

**Figure 8 ijms-20-04732-f008:**
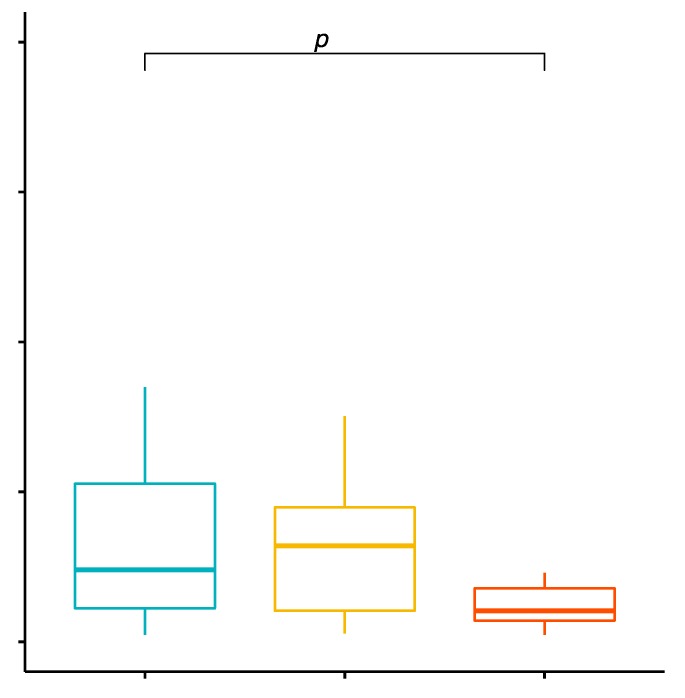
Percentage of CD19+/TLR-4+ B lymphocytes (among CD19+ cells; %) in patients with GD before treatment, after euthyroidism and in the control group.

**Figure 9 ijms-20-04732-f009:**
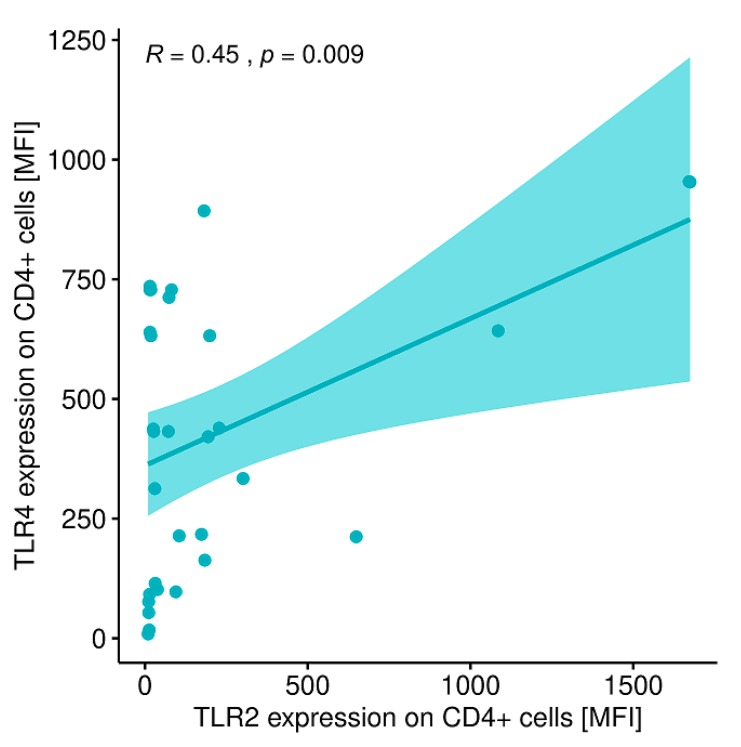
Correlation between the expression level of TLR-2 on CD4+ T lymphocytes (MFI) and the expression level (MFI) of TLR-4 on CD4+ T lymphocytes (MFI) in patients with GD before treatment. MFI, mean fluorescence intensity.

**Figure 10 ijms-20-04732-f010:**
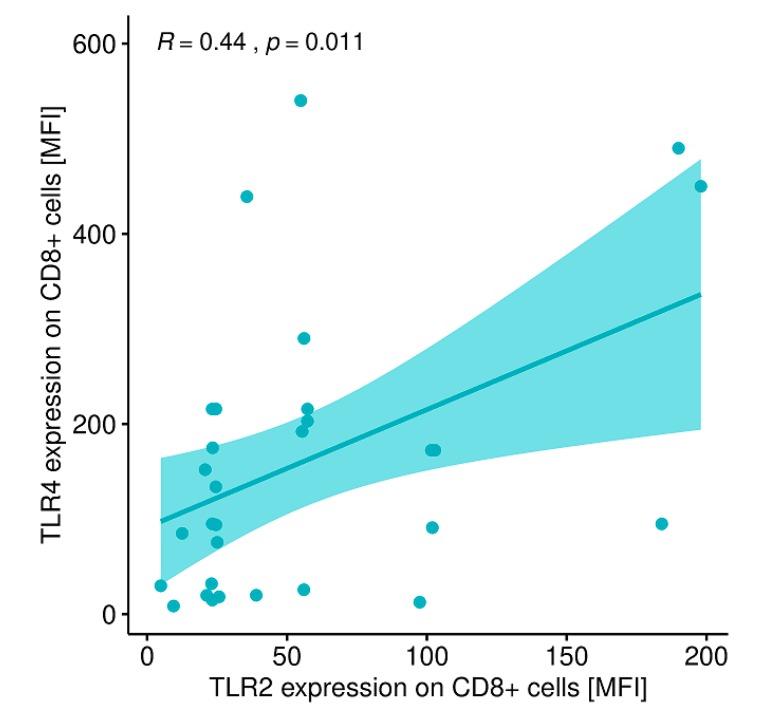
Correlation between the expression level of TLR-2 on CD8+ T lymphocytes (MFI) and the expression level of TLR-4 on CD8+ T lymphocytes (MFI). MFI, mean fluorescence intensity.

**Figure 11 ijms-20-04732-f011:**
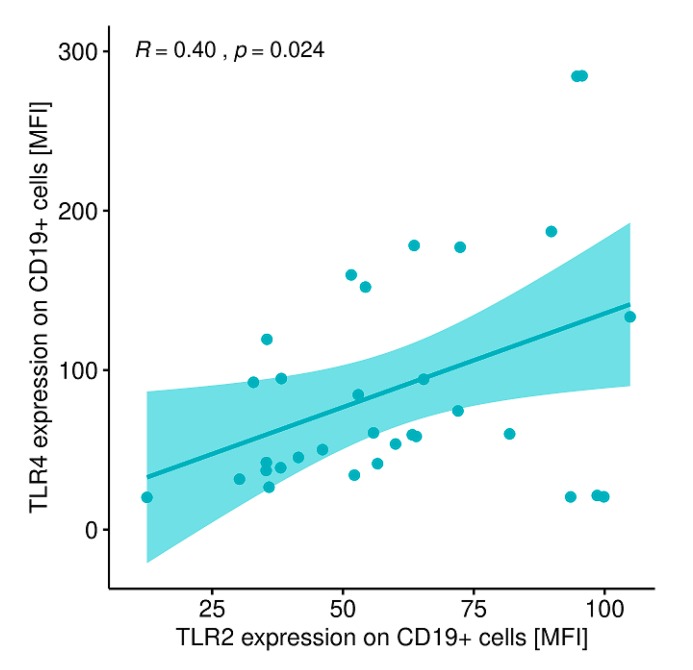
Correlation between the expression level of TLR-2 on CD19+ B lymphocytes (MFI) and the expression level of TLR-4 on CD19+ B lymphocytes (MFI). MFI, mean fluorescence intensity.

**Figure 12 ijms-20-04732-f012:**
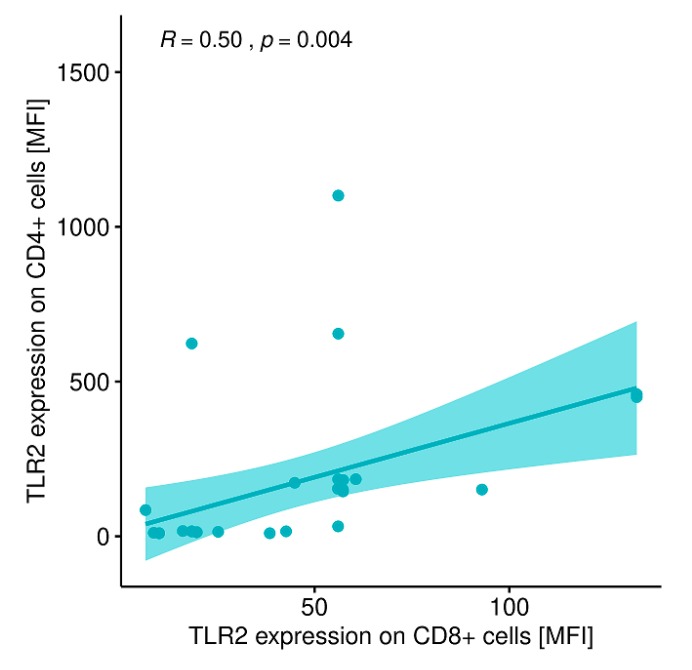
Correlation between the expression of TLR-2 on CD8+ T lymphocytes (MFI) and the expression level of TLR-2 on CD4+ T lymphocytes (MFI) in patients with GD after obtaining euthyroidism. MFI, mean fluorescence intensity.

**Figure 13 ijms-20-04732-f013:**
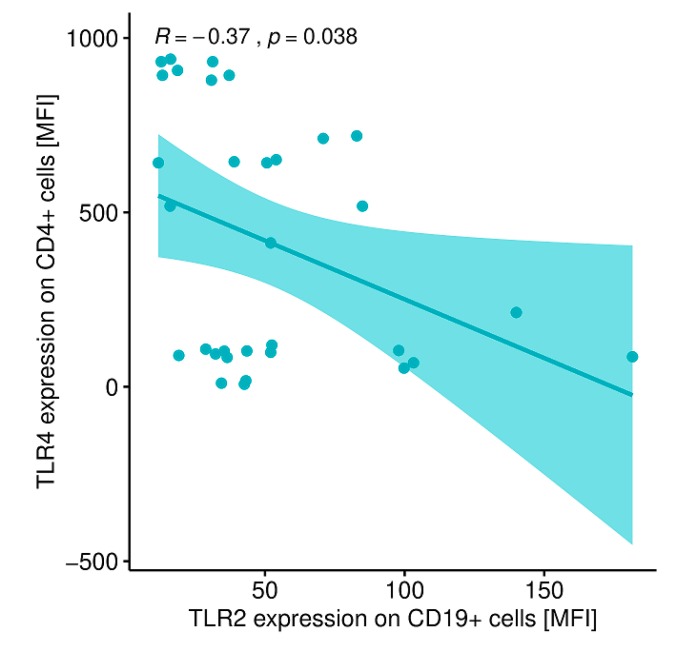
Correlation between the expression level of TLR-2 on CD19+ B lymphocytes (MFI) and the expression level of TLR-4 on CD4+ T lymphocytes (MFI) in patients with GD after obtaining euthyroidism. MFI, mean fluorescence intensity.

**Figure 14 ijms-20-04732-f014:**
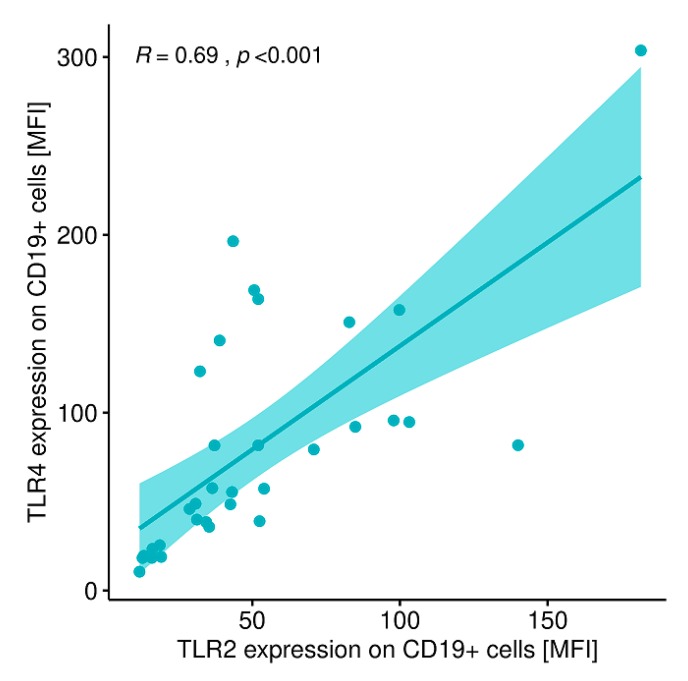
Correlation between the expression level of TLR-2 on CD19+ B lymphocytes (MFI) and the expression level of TLR-4 on CD19+ B lymphocytes (MFI) in patients with GD after obtaining euthyroidism. MFI, mean fluorescence intensity.

**Figure 15 ijms-20-04732-f015:**
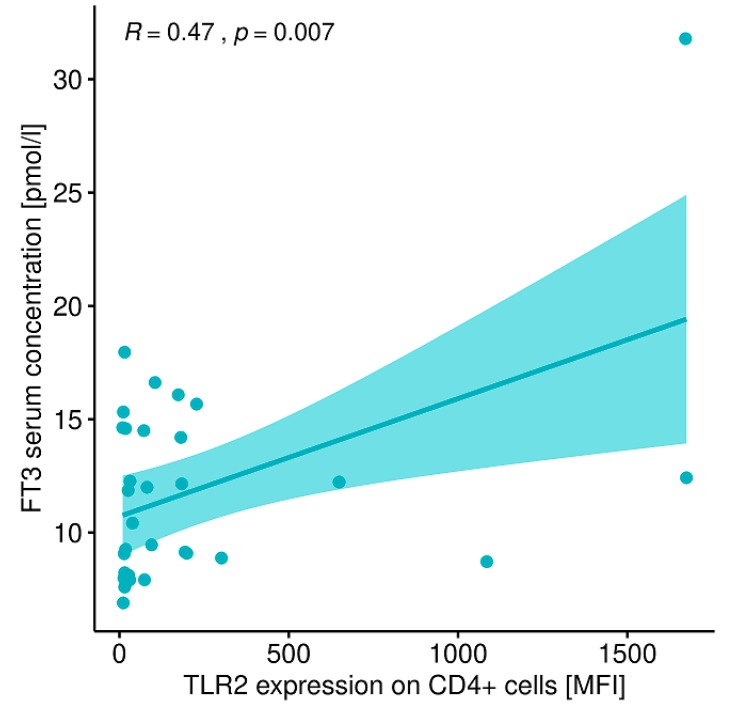
Correlation between TLR-2 expression level on CD4+ T lymphocytes (MFI) and FT3 (pmol /L) concentration in patients with GD before treatment. MFI, mean fluorescence intensity.

**Figure 16 ijms-20-04732-f016:**
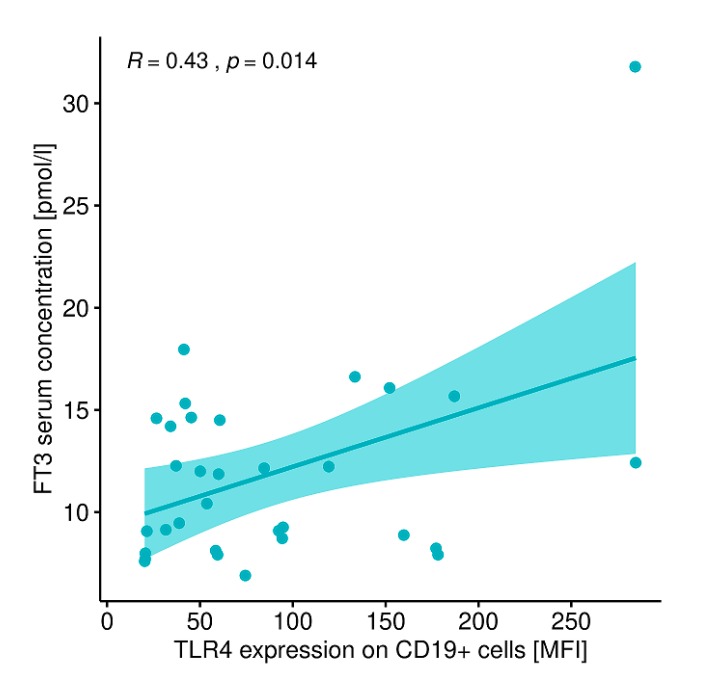
Correlation between the expression level of TLR-4 on CD19+ B lymphocytes (MFI) and the concentration of FT3 (pmol/L) in patients with GD before treatment. MFI, mean fluorescence intensity.

**Figure 17 ijms-20-04732-f017:**
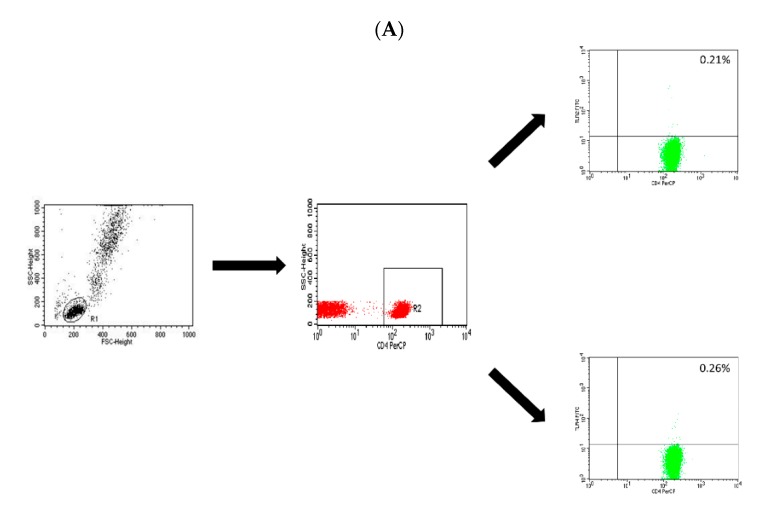
Dot-plots showing the percentage of T CD4+/TLR-2+ cells and T CD4+/TLR-4+ cells (**A**), CD8+/TLR-2+ T cells and CD8+/TLR-4+ T cells (**B**) and CD19+/TLR-2+ B cells and CD19+/TLR-4+ B cells (**C**).

**Table 1 ijms-20-04732-t001:** Characteristics of the study group (*n* = 32).

Age (years)	Mean ± SD	41.00 ± 16.21
	Median	36.00
	Min.	22.00
	Max.	95.00
Signs and symptoms duration from the time of diagnosis until the implementation of thyrostatic therapy (months)	Mean ± SD	2.34 ± 2.01
	Median	2.00
	Min.	0.00
	Max.	8.00
TRAb (U/l)	Mean ± SD	12.17 ± 9.69
	Median	9.75
	Min.	1.70
	Max.	39.40
Anti-TPO antibodies (U/mL)	Mean ± SD	1871.47 ± 2691.30
	Median	1237.50
	Min.	13.70
	Max.	14753.00
Anti-TG antibodies (U/mL)	Mean ± SD	250.43 ± 325.95
	Median	129.00
	Min.	10.00
	Max.	1360.00
Thyroid gland volume (mL)	Mean ± SD	21.20 ± 7.34
	Median	20.35
	Min.	13.30
	Max.	47.60
Methimazole daily dosage (mg)	Mean ± SD	34.53 ± 7.66
	Median	40.00
	Min.	20.00
	Max.	45.00
Time of thyrostatic therapy until receiving euthyroidism (methimazole treatment; days)	Mean ± SD	45.75 ± 9.38
	Median	43.00
	Min.	31.00
	Max.	74.00
Relapse of hyperthyroidism	Yes—number of patients (%)	10 (31.25%)
	No—number of patients (%)	22 (68.75%)
Time elapsed between receiving euthyroidism and relapse (months)	Mean ± SD	22.50 ± 11.00
	Median	24.50
	Min.	6.00
	Max.	35.00

**Table 2 ijms-20-04732-t002:** Comparison of the study (*n* = 32) and control (*n* = 20) groups.

		GD Patients before Treatment	GD Patients after Receiving Euthyroidism	Control Group	GD Patients before Treatment vs. GD Patients after Receiving Euthyroidism	GD Patients before Treatment vs. Control Group	GD Patients after Receiving Euthyroidism vs. Control Group
TSH concentration in the peripheral blood (mU/L)	Mean ± SD	0.01 ± 0.02	0.15 ± 0.34	2.71 ± 0.81	NS	*p* < 0.0001	*p* < 0.0001
	Median	0.01	0.02	2.71			
	Min.	0.0010	0.0020	0.92			
	Max.	0.09	1.33	3.98			
FT3 concentration in the peripheral blood (pmol/L)	Mean ± SD	11.90 ± 4.82	5.42 ± 1.24	4.97 ± 0.74	*p* < 0.0001	*p* < 0.0001	NS
	Median	11.14	4.89	4.88			
	Min.	6.90	3.78	3.60			
	Max.	31.79	7.66	6.40			
FT4 concentration in the peripheral blood (pmol/L)	Mean ± SD	35.01 ± 15.26	16.93 ± 2.78	17.27 ± 1.99	*p* < 0.0001	*p* < 0.0001	NS
	Median	30.95	16.34	17.04			
	Min.	19.05	12.23	14.00			
	Max.	82.22	22.39	21.47			
Lymphocytosis (10^3^/mm^3^)	Mean ± SD	1.85 ± 0.45	2.58 ± 0.43	2.35 ± 0.59	NS	NS	NS
	Median	1.75	2.57	2.37			
	Min.	1.22	1.75	1.39			
	Max.	2.87	3.34	3.38			
T CD3+/CD4+ lymphocytes (%)	Mean ± SD	41.96 ± 1.15	41.56 ± 1.27	42.5 ± 1.33	NS	NS	NS
	Median	42.47	41.19	42.45			
	Min.	40.63	40.54	41.06			
	Max.	43.99	44.19	44.13			
T CD3+/CD8+ lymphocytes (%)	Mean ± SD	30.79 ± 1.07	29.42 ± 0.46	30.58 ± 1.23	NS	NS	NS
	Median	30.61	29.47	30.29			
	Min.	29.35	28.9	29.45			
	Max.	33.08	30.09	33.17			
B CD19+ lymphocytes (%)	Mean ± SD	11.17 ± 1.02	10.26 ± 2.61	10.59 ± 1.89	NS	NS	NS
	Median	11.61	10.31	10.5			
	Min.	9.46	6.04	7.62			
	Max.	12.23	14.52	13.9			
SPINA-GD	Mean ± SD	33.91 ± 12.85	30.49 ± 8.58	26.75 ± 3.61	NS	*p* = 0.007	NS
	Median	33.09	30.13	27.01			
	Min.	12.79	16.85	19.61			
	Max.	80.66	45.64	33.31			
SPINA-GT	Mean ± SD	2336.39 ± 2665.83	428.71 ± 589.93	2.778 ± 0.55	*p* < 0.0001	*p* < 0.0001	*p* < 0.0001
	Median	893.37	183.838	2.58			
	Min.	59.71	3.27	2.11			
	Max.	8388.75	2242.32	4.48			
JTI	Mean ± SD	−0.34 ± 2.11	−1.40 ± 1.74	3.27 ± 0.53	*p* = 0.020	*p* < 0.0001	*p* < 0.0001
	Median	−1.00	−1.89	3.37			
	Min.	−2.96	−3.52	1.91			
	Max.	5.76	2.66	4.24			

NS—not statistically significant.

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
