# Peer review of "Toll-Like Receptors-2 and -4 in Graves’ Disease—Key Players or Bystanders?"

_ijms, 2019, doi:10.3390/ijms20194732_

Round 1
Reviewer 1 Report
The paper by Polak et al. is a very interesting report on the role of toll-like receptors -2 and -4 in the pathogenesis and course of Graves’ disease. The importance of these receptors in the pathogenesis of autoimmune diseases is currently widely studied, and the authors were the first to examine this issue in relation to Graves' disease.
However, several issues required revision and improvement:
The most important weakness of the study is the small size of the groups. The incidence of Graves’ disease is rather high in Polish population, while the study group included only 32 people. The authors should analyze the parameters for a larger number of patients, or – if for some reason it is impossible – they should refer to this limitation in the manuscript and explain it. Several sentences require significant language correction, e.g. lines 268, 280, 308-310, 311 “in their work” by Peng”, line 324 , 375 “5 ml patients”, Please, do not use capital letter after “et al.” e.g. lines 272, 316. Please change the “conscious written consent” to “written informed consent” line 377 Please correct citations e.g. line 327. Author Contributions should be prepared according to the instruction for authors, based on the information introduced during the submission process. The paragraphs regarding Appendix A and B should probably be removed – lines 451-458. The introduction to references (lines 460-470) should be removed from the manuscript. It is just the manual for a manuscript preparation. Please correct the Abbreviations section. There are several abbreviations that do not appear in the manuscript and many of these actually present are not listed.Author Response
REVIEWER 1
The most important weakness of the study is the small size of the groups. The incidence of Graves’ disease is rather high in Polish population, while the study group included only 32 people. The authors should analyze the parameters for a larger number of patients, or – if for some reason it is impossible – they should refer to this limitation in the manuscript and explain it.
Our study had a proof-of-concept nature, hence the small sample. We agree that a larger study could provide more robust conclusions, and we mention this limitation in the discussion.
Several sentences require significant language correction, e.g. lines 268, 280, 308-310, 311 “in their work” by Peng”, line 324 , 375 “5 ml patients”,
(Line 268) Corrected sentence: Most studies have concerned intra-cellular TLRs, such as TLR-3, TLR-7, and TLR-9, but recent studies have also shown that cell-surface TLRs, especially TLR-2 and TLR-4, play an equally important role in the development of autoimmune diseases [16]; however, relevant studies have not been conducted so far in women with GD.
(Line 280) Corrected sentence: Studies by Komatsud et al. indicated that TLR-2 mRNA levels significantly increased in the peripheral blood of SLE patients compared to the control group, whereas Kirchner et al. demonstrated that the level of TLR-4 expression on CD14 + monocytes was significantly lower in patients with SLE compared to the control group [19, 20].
(Lines 308-310, 311) Corrected sentence: Our research showed that the mean percentages of CD4 + / TLR-2 + T cells, CD8 + / TLR-2 + T cells, and CD19 + / TLR 2+ CD8 cells in peripheral blood were significantly greater in patients with GD than in the control group. Similar results were obtained by Peng et al.
(Line 324) Corrected sentence: … makes it impossible to compare our results with previous studies…
(Line 375) Corrected sentence: The study material was 5 ml of peripheral blood, collected in heparin-coated tubes to obtain peripheral blood cells, taken during routine laboratory tests.
Please, do not use capital letter after “et al.” e.g. lines 272, 316. Please change the “conscious written consent” to “written informed consent” line 377 Please correct citations e.g. line 327. Author Contributions should be prepared according to the instruction for authors, based on the information introduced during the submission process. The paragraphs regarding Appendix A and B should probably be removed – lines 451-458. The introduction to references (lines 460-470) should be removed from the manuscript. It is just the manual for a manuscript preparation.
We corrected all these errors accordingly.
Please correct the Abbreviations section. There are several abbreviations that do not appear in the manuscript and many of these actually present are not listed.
The list of abbreviations was updated.
Reviewer 2 Report
In their manuscript with title “Toll-like receptors -2 and -4 in Graves’ disease - key players or bystanders?” Agnieszka Polak et al. describe the result of a study in female patients with Graves’ disease (GD) and healthy volunteers. They observed that the expression of certain toll-like receptors (TLR2 and TLR4) in CD4+ and CD8+ T cells was increased in GD, that the expression level correlated to serum FT3 concentration and that it decreased after the patients arrived at euthyroidism.
This is a well-planned study addressing an important question since the immune pathology of autoimmune thyroiditis and especially of Graves’ disease is still understudied and under-recognized. Several minor issues make the manuscript unacceptable in its current form, however.
First of all, the link between immune function and the endocrine phenotype is only sparsely addressed. The authors extensively describe the relation between TLR expression and FT3 concentration, but the reader learns nearly nothing about the relation to TSH and FT4 concentrations. Was there any correlation? A missing correlation would be worth reporting, too. Additionally, the authors should report (existent or missing) correlations to thyroid’s secretory capacity (SPINA-GT), step-up deiodinase activity (SPINA-GD) and Jostel’s TSH index (JTI), since these reconstructed structure parameters may represent important pathophysiological links. Correlation of TLR expression to SPINA-GD would suggest the TSH-T3 shunt to be activated by the autoimmune process, which would explain the still poorly understood high-T3 syndrome in GD. Additionally, the relation of JTI to the immune process would be interesting in the context of the still enigmatic involvement of the ultrashort-loop feedback control of TSH secretion in GD (see papers by Leon Brokken, Mark Prummel and Wilmar Wiersinga for reference). Additionally, correlations to thyroid volume and TRAb titres would be of interest. Generally, all correlations that are reported for FT3 should also be reported for TSH, FT4, SPINA-GT, SPINA-GD, JTI, thyroid volume and TRAb (of course and for sake of brevity, this doesn’t necessarily imply the inclusion of additional figures for all of these correlations). Statistical moments of SPINA-GT, SPINA-GD and JTI should also be reported in table 2.
Did some of the subjects suffer from Graves’ ophthalmopathy, and if so, how many? Was there any correlation of Hertel exophthalmometry to immunological parameters?
Additionally, the authors should try to explain the negative correlation between TLR2 expression on CD19+ cells and TLR4 expression on CD4+ cells in the discussion.
It was only briefly mentioned that the subjects with GD received thyrostatic treatment (line 371). Which specific drug(s) was/were used and what was the dosage range? This may be especially important since carbimazole is believed to have an immunosuppressive effect (see e.g. papers by McGregor et al. 1982 and by Wilson et al. 1988).
The second part of the title “key players or bystanders?” promises an interesting ventilation of this chicken-and-egg problem, but this discussion is totally missing. The paper would benefit from addressing this question.
The quality of the English language is rather good. Some minor vocabular issues should be resolved, however. This includes the usage of the word “euthyreosis”, which is uncommon in the English language. It should be replaced by the established term “euthyroidism”, wherever it occurs. The words “thyreocytes” and “thyreostatic” should be replaced by “thyrocytes” and “thyrostatic”, respectively. In lines 198, 205, 215, 222, 236 and 241 the definite article “The” should be replaced by the indefinite article “A” (since the respective correlations hadn’t been previously introduced). Likewise, “the” should be replaced by “a” in line 276. The words “most studies” are doubled in line 268. The fragment of the sentence in line 270 is incomprehensible. This also applies to line 316. “Demonstrated” should be in lower case in lines 272 and 281. In line 428, “inclusion” should be replaced by “introduction” and “after euthryreosis” should be replaced by “after arrival at euthyroidism”.
In places, the authors didn’t remove boilerplate blocks from the text templated provided by the publisher. This applies to lines 19 (“(optional; include country code)”), lines 452 to 457, and lines 460 to 470.
Figures 17 A, 17 B and 17 C are too small to be readable.
Author Response
REVIEWER 2
The authors extensively describe the relation between TLR expression and FT3 concentration, but the reader learns nearly nothing about the relation to TSH and FT4 concentrations. Was there any correlation? A missing correlation would be worth reporting, too. Additionally, the authors should report (existent or missing) correlations to thyroid’s secretory capacity (SPINA-GT), step-up deiodinase activity (SPINA-GD) and Jostel’s TSH index (JTI), since these reconstructed structure parameters may represent important pathophysiological links. Correlation of TLR expression to SPINA-GD would suggest the TSH-T3 shunt to be activated by the autoimmune process, which would explain the still poorly understood high-T3 syndrome in GD. Additionally, the relation of JTI to the immune process would be interesting in the context of the still enigmatic involvement of the ultrashort-loop feedback control of TSH secretion in GD (see papers by Leon Brokken, Mark Prummel and Wilmar Wiersinga for reference). Additionally, correlations to thyroid volume and TRAb titres would be of interest. Generally, all correlations that are reported for FT3 should also be reported for TSH, FT4, SPINA-GT, SPINA-GD, JTI, thyroid volume and TRAb (of course and for sake of brevity, this doesn’t necessarily imply the inclusion of additional figures for all of these correlations). Statistical moments of SPINA-GT, SPINA-GD and JTI should also be reported in table 2.
Thank you for these insights. As suggested, we calculated the additional thyroid parameters (SPINA-GD, SPINA-GT, JTI – reported in Table 2 and section 2.13) and additional correlation coefficients (reported in section 2.12 – there were no significant correlations). In the methods, we specify how these new variables were calculated. In the discussion, we comments on these findings, as follows:
“In our study, SPINA-GD (sum activity of peripheral deiodinases) was increased in patients with GD before treatment, and it decreased to near normal values when the patients were euthyroid. SPINA-GD correlates with the FT4 to FT3 conversion rate; thus, an increased SPINA-GD may be related to the high-FT3 syndrome in GD. As one could expect, the thyroid secretory capacity (SPINA-GT) was markedly increased in patients with GD before treatment, and it decreased, but did not normalize, after thyrostatic treatment. The mean JTI was negative in patients with GD, which was due to very low TSH values [JTI = ln (TSH) + 0.1345 X FT4]. Interestingly, the JTI decreased further after thyrostatic treatment, likely because FT4 levels normalize faster than TSH in GD.
Did some of the subjects suffer from Graves’ ophthalmopathy, and if so, how many? Was there any correlation of Hertel exophthalmometry to immunological parameters?
None of the patients had exophthalmos, and we did not use Hertel exophthalmometry.
Additionally, the authors should try to explain the negative correlation between TLR2 expression on CD19+ cells and TLR4 expression on CD4+ cells in the discussion.
After a thorough literature search, we could not find any feasible explanation of this finding; we express our lack of knowledge in the discussion:
“It is difficult to explain the negative correlation between TLR2 expression on CD19+ cells and TLR4 expression on CD4+ cells. Perhaps, there is a negative feedback loop that controls the expressions of TLR-4 and TLR-2, but we are not aware of any previous evidence that could support this hypothesis.”
It was only briefly mentioned that the subjects with GD received thyrostatic treatment (line 371). Which specific drug(s) was/were used and what was the dosage range? This may be especially important since carbimazole is believed to have an immunosuppressive effect.
In Table 2, we report the mean dose of methimazole (thiamzaole), which was the only thyrostatic medications used.
The second part of the title “key players or bystanders?” promises an interesting ventilation of this chicken-and-egg problem, but this discussion is totally missing. The paper would benefit from addressing this question.
In the study limitations, we admit that our observational study cannot answer this question. We write:
“Moreover, although our observational study points to the importance of TLRs in AITD, it cannot answer whether TLRs are key players or by-standers in the pathogenesis of GD. The role of TLRs in AITD should be further investigated in preclinical studies that enable TLR blockage or stimulation. Similarly, large studies among patients with GD could help reveal any link between TLRs expression and relevant clinical outcomes”.
The usage of the word “euthyreosis”, which is uncommon in the English language. It should be replaced by the established term “euthyroidism”, wherever it occurs. The words “thyreocytes” and “thyreostatic” should be replaced by “thyrocytes” and “thyrostatic”, respectively. In lines 198, 205, 215, 222, 236 and 241 the definite article “The” should be replaced by the indefinite article “A” (since the respective correlations hadn’t been previously introduced). Likewise, “the” should be replaced by “a” in line 276.
These errors were corrected accordingly.
The words “most studies” are doubled in line 268. The fragment of the sentence in line 270 is incomprehensible. This also applies to line 316.
Line (268) Corrected sentence: Most studies have concerned intra-cellular TLRs, such as TLR-3, TLR-7, and TLR-9, but recent studies have also shown that cell-surface TLRs, especially TLR-2 and TLR-4, play an equally important role in the development of autoimmune diseases [16]; however, relevant studies have not been conducted so far in women with GD.
Line (316) Corrected sentence: They observed, among others things, a markedly increased TLR-2 expression in the peripheral blood of patients with newly diagnosed GD or with Hashimoto thyroiditis compared to the control group
“Demonstrated” should be in lower case in lines 272 and 281. In line 428, “inclusion” should be replaced by “introduction” and “after euthryreosis” should be replaced by “after arrival at euthyroidism”.
These errors were corrected.
In places, the authors didn’t remove boilerplate blocks from the text templated provided by the publisher. This applies to lines 19 (“(optional; include country code)”), lines 452 to 457, and lines 460 to 470.
These were removed.
Figures 17 A, 17 B and 17 C are too small to be readable.
We replaced these figures.
We hope that you will find the manuscript much improved and suitable for publication in the “International Journal of Molecular Sciences”. I will be glad to give you any further information.
Kind regards,
Ewelina Grywalska on behalf of all authors.